# Socioeconomic, Demographic, and Environmental Factors May Inform Malaria Intervention Prioritization in Urban Nigeria

**DOI:** 10.3390/ijerph21010078

**Published:** 2024-01-10

**Authors:** Chilochibi Chiziba, Laina D. Mercer, Ousmane Diallo, Amelia Bertozzi-Villa, Daniel J. Weiss, Jaline Gerardin, Ifeoma D. Ozodiegwu

**Affiliations:** 1Department of Preventive Medicine and Institute for Global Health, Northwestern University, Chicago, IL 60611, USA; 2PATH, Seattle, WA 98121, USA; 3Institute for Disease Modeling, Seattle, WA 98005, USA; 4Telethon Kids Institute, Nedlands, WA 6009, Australia; 5Faculty of Health Sciences, Curtin University, Bently, WA 6102, Australia; 6Department of Health Informatics and Data Science, Loyola University, Health Sciences Campus, Maywood, IL 60153, USA

**Keywords:** urbanization, malaria, risk factors, cities, Nigeria

## Abstract

Urban population growth in Nigeria may exceed the availability of affordable housing and basic services, resulting in living conditions conducive to vector breeding and heterogeneous malaria transmission. Understanding the link between community-level factors and urban malaria transmission informs targeted interventions. We analyzed Demographic and Health Survey Program cluster-level data, alongside geospatial covariates, to describe variations in malaria prevalence in children under 5 years of age. Univariate and multivariable models explored the relationship between malaria test positivity rates at the cluster level and community-level factors. Generally, malaria test positivity rates in urban areas are low and declining. The factors that best predicted malaria test positivity rates within a multivariable model were post-primary education, wealth quintiles, population density, access to improved housing, child fever treatment-seeking, precipitation, and enhanced vegetation index. Malaria transmission in urban areas will likely be reduced by addressing socioeconomic and environmental factors that promote exposure to disease vectors. Enhanced regional surveillance systems in Nigeria can provide detailed data to further refine our understanding of these factors in relation to malaria transmission.

## 1. Introduction

Nigeria accounts for 27 percent of all global malaria cases and 31 percent of global malaria deaths, making it the greatest contributor to the global malaria burden [1]. Underlying Nigeria’s malaria burden are spatial and temporal differences in malaria risk driven by diverse factors, including ecological and climatic factors, intervention histories, health system factors, land use practices, and urbanization trends [2,3,4]. Notably, Nigeria is among the top three countries expected to contribute to nearly one-third of the world’s urban population growth between 2018 and 2050 [5]. In 2018, approximately half of Nigeria’s 200 million population resided in urban areas, and this proportion is projected to surge to 70 percent by 2050 [5]. These trends in urban population expansion raise concerns about the concentration of Nigeria’s malaria burden in urban areas. It is even more concerning that these population shifts are occurring in an atmosphere of declines in donor funding for malaria interventions [6]. Given the confluence of rapid urbanization in Nigeria and funding limitations, the scientific community and policymakers are increasingly interested in understanding how urbanization-related factors may impact malaria transmission to inform improved allocation of available resources.

An examination of data compiled from across Africa, coupled with in-depth investigations into the consequences of urbanization in Brazzaville, Republic of Congo, illuminates several key urban community-level factors that influence malaria transmission in cities [7,8,9]. Infrastructure development, heightened population density, and improved access to healthcare services are factors that are generally expected to reduce malaria transmission within urban settings. However, the rapid and unplanned urbanization characterized by the establishment of farms within urban neighborhoods, as observed in Cotonou, Benin [10], and the adaptation of the *Anopheles gambiae* sensu lato mosquito to polluted waters, witnessed in Ghana [11], Cameroon [12], and Sudan [13], pose significant threats that can elevate the risk of malaria transmission in cities. Furthermore, reports of the proliferation of *Anopheles stephensi*, a major malaria vector well-suited to urban environments, in East Africa and Nigeria [6], add to these concerns. In addition to these factors, the increasing mobility of individuals between urban areas and other settings as documented in Uganda [14] and Burkina Faso [15] is among the factors that could perpetuate malaria transmission within urban spaces.

The existing online research literature specific to Nigeria indicates that community determinants of urban malaria transmission risk encompass factors such as proximity to water bodies, travel to rural areas, environmental hygiene practices, and housing quality [16,17,18,19,20]. For instance, a study conducted by Awosolu and colleagues, focusing on patients receiving care at two hospitals in Ibadan, revealed that factors, such as residing within a 1 km distance from streams and recent travel to a rural area, were statistically significant risk factors for malaria infection. Similarly, findings from a cross-sectional survey conducted in an urban town in Nigeria’s southwest region indicated that the types of windows and environmental hygiene practices significantly predicted malaria prevalence within households [20]. While these studies provide valuable insights, their limited sample sizes and focus on individual cities may constrain their applicability for making informed decisions regarding appropriate malaria interventions. Therefore, complementary research endeavors are essential to provide a comprehensive overview of key factors associated with malaria transmission across various cities in Nigeria.

Georeferenced survey data, such as the Malaria Indicator Surveys (MIS) and Demographic and Health surveys (DHS), alongside modeled geospatial data, offer valuable tools for comprehending the risk of malaria transmission in urban areas. Unlike routine surveillance systems, which typically lack individual-level georeferenced data on malaria infections and mostly include individuals who seek care in public healthcare institutions [21,22], the MIS and DHS collect georeferenced data on an individual’s infection status and risk factors in both urban and rural areas [23]. Data from individuals are organized by clusters or enumeration areas, representing the aggregation of households. The geographic coordinates from the MIS/DHS pertain to point data collected at the centroids of clusters. These clusters ensure comprehensive coverage of the administrative unit being sampled, with data collected through a random selection during household surveys. Augmented by geospatial covariates, these survey datasets facilitate the examination of associations with potential correlates of malaria infections within urban areas.

A notable limitation of using the MIS/DHS data is the restriction of malaria testing to children under five years of age, coupled with the absence of detailed information about the urban extent of clusters and their corresponding cities. Nonetheless, it is important to emphasize that children under five are a key demographic for malaria prevention and control efforts, regardless of the intensity of transmission and seasonality [24]. This underscores the relevance of our analysis. For the second limitation, this study’s significance is maintained because the cluster centroids are situated within areas classified as urban by local authorities. This alignment enhances the potential for policymakers to accept and act upon the results of the analysis.

In light of these data limitations and the imperative to comprehend malaria transmission risk in Nigerian cities, this study embarked on an analysis with the following objectives: (1) describe the magnitude and variations in malaria prevalence among children under the age of five years (U5), analyzed at both the cluster and geopolitical level in individual DHS/MIS surveys, (2) identify factors that predict the U5 malaria test positivity rates, and (3) construct model effect plots that elucidate the associations between covariates and the U5 malaria test positivity rates. The unadjusted effect plots reveal correlations between dependent and independent variables, providing insights into potential thresholds for intervention prioritization. Meanwhile, the adjusted effects illuminate the independent contributions of covariates, crucial for understanding their broader public health implications.

## 2. Materials and Methods

### 2.1. Data

Cluster-level data from the 2010, 2015, and 2021 MIS, the 2018 DHS, and publicly available geospatial malaria covariate data were used for this study (refer to Table 1 for references). The DHS program conducts complex multistage surveys to gather and disseminate accurate, nationally representative data on health and population in over 90 countries [23]. Data collection occurs at the individual level following probability proportional to size sampling of clusters, geocoded at their centroids, and a random selection of households within these clusters [23]. For this study, only clusters classified as being situated in urban areas were retained. Figure 1 provides an overview of the DHS/MIS sampling framework. It is important to note that the DHS program intentionally displaced the GPS coordinates of the centroid of urban clusters by 0 to 2 km to ensure the confidentiality of survey participants [25]. However, the displacement is performed in such a manner that each cluster remains within the state-level boundaries. Given that the DHS exclusively collects data on children aged 6 to 59 months, the number of positive malaria tests by microscopy and the testing sample population for this age group were summarized for each cluster. The selection of covariates was guided by the relevant research literature, which highlights socioeconomic, demographic, behavioral, accessibility, and environmental factors as potential explanatory variables for malaria transmission [2,4,15,26,27,28,29]. A detailed description of all 29 considered covariates considered can be found in Table 1. To account for the distance displacement of MIS and DHS clusters when calculating values for geospatial covariates, raster values were aggregated across buffers of up to 4 km around each MIS and DHS cluster.

### 2.2. Descriptive Analysis and Covariate Selection

To comprehensively grasp the magnitude and distribution patterns of malaria prevalence in children under the age of five (U5), a descriptive analysis was conducted on their test positivity rates. This rate, calculated as the number of U5 children with positive malaria tests divided by the total number of U5 children tested for malaria in each cluster, was mapped by the state and further compared across different months and years of the survey, as well as the geopolitical region. Following this analysis, covariates were summarized to assess their distribution and identify potential sources of information bias. Pearson correlation coefficients were calculated within thematic groups to identify any strong correlations among the covariates. The ggplot2 package in R [30] was utilized to fit a Poisson regression model, allowing for the visual examination of bivariate relationships between covariates and malaria test positivity rates. Non-linear relationships with covariates were estimated using natural cubic splines from the splines package in R [31]. In instances where two covariates exhibited a correlation coefficient of 60% or higher, the one displaying a weaker visual relationship with the malaria test positivity rate, along with wider confidence bands, was excluded from the subsequent multivariable analysis.

### 2.3. Multivariable Modeling

A model of the U5 malaria test positivity rate was constructed to identify predictive factors and generate effect plots. For this purpose, multivariable generalized linear models were employed, organized by thematic group and a combination of variables across thematic group. The glmmTMB package was employed, considering modeling-related factors such as zero inflation, temporal dynamics, and spatial dependence [32]. The dependent variable in these models represented the count of positive malaria tests among U5 children, adjusted for the total number of U5 children tested for malaria in each cluster using an offset term. Similar to the descriptive models previously discussed, we accounted for non-linear relationships by employing natural cubic splines. Temporal dependencies were addressed by incorporating the survey month and year for each cluster into the model, while spatial dependencies were considered by including the geographical coordinates of each cluster. The Akaike Information Criterion (AIC) statistic was used to select the best predictive model of the malaria test positivity rate. Ultimately, the selected final model took the form of a zero-inflated Poisson model. This choice was made based on goodness-of-fit tests, including the Kolmogorov–Smirnov test, dispersion test, and outlier tests, which were conducted using the DHARMa package. DHARMa utilizes simulation-based methods to produce interpretable scaled residuals for fitted generalized linear mixed models [33]. The model equation can be written as follows:(1)$$\lambda_{tz}=E\left(count|\mu \left(t\right),\;\eta \left(z\right),\;NSZ\right)=\exp\left(\beta_{0}+\beta_{1}h_{1}{\left(X_{1}\right)}^{T}+\cdots +\beta_{n}h_{n}{\left(X_{n}\right)}^{T}+\mu \left(t\right)+\eta \left(z\right)+w\right),$$
(2)$$\;logit\left(p\right)=\beta_{0}^{\left(zi\right)}\;,\;$$
where each of the $\beta_{j}$ are vectors of coefficients multiplying their associated vector natural spline basis functions; $h_{j}$ and $\beta_{0}$ represent the model intercept; $X_{i\ldots n}$ are covariate values; $\mu \left(t\right)$ is a cluster-specific stationary autoregressive (1) process (type of autoregressive model) for modeling temporal dependence by month and year of the survey; t represents each study cluster; $\eta \left(z\right)$ is the spatial Matern process for modeling spatial random effects using each cluster coordinate z; *NSZ* is the event non-structural zero; and w represents the offset term, which is the number of children 6–59 months tested for malaria. The zero components were modeled with the equation in (2), with p representing the probability of observing zero counts. Due to the large computational power required to generate cubic plots from complex models, unadjusted and adjusted effect plots for the final model covariates were produced and described using linear splines. All code written in support of this manuscript is available via this doi: https://zenodo.org/records/10426210 (accessed on 22 December 2023).

**Table 1 ijerph-21-00078-t001:** Variable names, definitions, and sources for a selection of cluster-level variables considered for modeling.

Variable Name	Variable Definition	Source
Dependent variable		
Number of U5 positive malaria tests	The number of positive malaria tests by microscopy among children 6–59 months old aggregated per cluster	MIS and DHS [23,34,35,36]
Explanatory variables by thematic group		
Socioeconomic factors		
1. % with post-primary education	Percentage (%) of women in each cluster with secondary or higher educational attainment	MIS and DHS [23,34,35,36]
2. % in the rich wealth quintiles	% of the cluster population in the rich and richest wealth quintiles. Wealth quintiles were constructed using various indicators of household living standards [36]	MIS and DHS [23,34,35,36]
3. % in homes with improved flooring	% of the cluster population living in homes with improved flooring (finished floors, parquet or polished wood, ceramic tiles, cement, and carpet)	MIS and DHS [23,34,35,36]
4. % in homes with a metal or zinc roof	% of the cluster population living in homes with a metal or zinc roof	MIS and DHS [23,34,35,36]
5. % in homes with an improved wall type	% of the cluster population living in homes with an improved wall type (finished wall, cement, bricks, cement blocks, covered adobe)	MIS and DHS [23,34,35,36]
6. % living in improved housing (2000)	Predicted % of the cluster population living in improved housing in 2000. Improved housing is defined as homes with improved water and sanitation, sufficient living area, and durable construction, according to Tusting et al. [37]	Malaria Atlas Project (MAP) [38,39]
7. % living in improved housing (2015)	Predicted % of the cluster population living in improved housing in 2015	MAP [37,40]
Demographic factors		
8. All-age population density	Estimated population density per cluster at the time of the 2010 and 2015 DHS/MIS surveys. Population density data for 2020 were extracted for the 2018 and 2021 surveys (UN World Population Prospects-Adjusted Population Density, v4.11). Unit is persons per square kilometer	Center for International Earth Science Information Network, Columbia University [39]
9. Population density, children five years and under	Estimated population density for children under the age of five in 2020. Unit is the number of children per square kilometer	Humanitarian Data Exchange [40]
10. % of pregnant women	% of pregnant women	MIS and DHS [23,34,35,36]
11. % of the female population	% of females per cluster	MIS and DHS [23,34,35,36]
12. Median household size	Median household size per cluster	MIS and DHS [23,34,35,36]
13. Median age	Median age per cluster	MIS and DHS [23,34,35,36]
14. State	State where the cluster is located	MIS and DHS [23,34,35,36]
15. Region	Geopolitical region where the cluster is located (there are six geopolitical regions in Nigeria: the northeast, northwest, north central, southeast, southwest, south south)	MIS and DHS [23,34,35,36]
Behavioral factors		
16. % of individuals using bednets	% of the cluster population that slept under a treated bednet the night before the survey	MIS and DHS [23,34,35,36]
17. % of children 6–59 months using bednets, among those tested by microscopy	% of children 6–59 months tested for malaria by microscopy that slept under a treated bednet the night before the survey	MIS and DHS [23,34,35,36]
18. % of U5 children that sought medical treatment for fever	% of children under the age of five that received medical treatment given that they had a fever or cough in the two weeks before the survey. Medical treatment must be received in the public sector or medical private sector, except for a pharmacy	MIS and DHS [23,34,35,36]
19. % of U5 children with a fever who received artemisinin combination therapy (ACT)	% of children under the age of five that received an ACT given that they had a fever	MIS and DHS [23,34,35,36]
Accessibility-related factors		
20. Motorized travel time to healthcare in minutes	Predicted travel time to healthcare facility in minutes in 2019	MAP [41]
Environmental factors		
21. Total precipitation (depth in meters)	Estimated total precipitation during survey month and year per cluster. Units measure the depth in meters. It is measured as the depth that the water would have if it were spread evenly over a grid box	European Center for Medium-Range Weather Forecasts (ECMWF), Climate Data Store [42]
22. Temperature (°C)	Estimated temperature of air at 2 m above the surface of land, sea, or inland waters in Celsius per cluster during the survey month	ECMWF, Climate DataStore [42]
23. Surface soil moisture (GSM)	The average estimated depth of water present in a specific soil layer beneath the surface is measured as gravimetric soil moisture (GSM) per cluster. GSM is the mass of water compared to the mass of solid materials per unit volume of soil	Goddard Earth Sciences Data and Information Services Center [43]
24. Distance to water bodies (meters)	Straight line distance to water bodies in meters	MAP (unpublished data)
25. Elevation (meters)	Cluster elevation above sea level in meters	Multi-Error-Removed Improved-Terrain DEM [44]
26. Enhanced vegetation index	Enhanced vegetation index for quantifying vegetation greenness in units of the spectral index	MAP gap filled EVI (communication with MAP)
Other adjustment variables		
1. Number of children tested	Number of children 6–59 months old tested for malaria per cluster	MIS and DHS [23,34,35,36]
2. Interview date	Year the DHS survey was conducted per cluster and survey month per cluster (some clusters were surveyed over a two-month period; the first interview month was used in those cases)	MIS and DHS [23,34,35,36]
3. Longitude and latitude	The longitude and latitude positions where the clusters were geolocated after displacement to protect participant confidentiality	MIS and DHS [23,34,35,36]

## 3. Results

### 3.1. Describing Variations in Malaria Prevalence among Children under the Age of Five Years in Urban Areas

#### 3.1.1. Sample Overview

A total of 988 clusters were sampled in 2010, 2015, 2018, and 2021 DHS and MIS surveys. The number of individuals surveyed in each cluster varied widely across all surveys, with ranges from 98 to 2949 in 2010, 167 to 2954 in 2015, 3 to 3471 in 2018, and 166 to 4765 in 2021. Malaria test results by microscopy were available for children 6–59 months in 972 of the 988 sampled clusters. The study dataset consisted of 81 clusters from the 2010 survey, 136 clusters from 2015, 560 clusters from 2018, and 195 clusters from 2021 (Figure 2A). On average, a higher number of children were tested per cluster in 2010, 2015, and 2021 compared to 2018 (Figure 2B). The 2010 clusters were sampled during the months of October, November, and December. The 2015 clusters were sampled in October and November. The 2018 clusters spanned from August to December, while the 2021 survey sampling took place in October, November, and December. Visualizing cluster centroids highlights the abundance of sampled clusters from the 2018 survey and that most clusters, 364 (37%) of the 972, were sampled in October, while the least number of clusters were sampled in August (Figure 2C,D).

#### 3.1.2. Low Malaria Test Positivity across the Majority of Urban Clusters

On average, ten children (mean = 10.2, standard deviation (SD) = 7.5) were tested per cluster. The distribution of the number of children that tested positive for malaria per cluster was predominantly skewed toward zero (Figure 3A), with a mean of 2.9 (SD = 2.8). The median test positivity rate was zero (interquartile range (IQR): 0.2). No child tested positive for malaria in roughly 49% (473) of the clusters. Stratifying clusters by year of survey revealed that the median test positivity rate was 0.1 for those surveyed in 2010 and 2015 and zero in 2018 and 2021, and that the test positivity rate declined over time (black line in Figure 3B denotes the median). Most clusters in Lagos (90%), Rivers (76%), Abia (75%), Akwa Ibom (75%), and Benue (75%) had a zero test positivity rate (Figure 3C). It is essential to note that the DHS 2018 report highlighted the exclusion of 11 Local Government Areas (LGAs) in Borno during the initial sampling phase due to security concerns [23]. This exclusion raises concerns about the representativeness of the test positivity rate distribution within this state, located in Nigeria’s northeast. At the regional level, 64% of the clusters in the south south geopolitical region had a zero test positivity rate, 63% of the clusters in the northeast, 61% in the north central, 50% in the southeast, 49% in the southwest, and 48% in the northwest (Figure 3D).

#### 3.1.3. Test Positivity Rates in Sampled Clusters Declined over Time

To assess whether the observed time-related declines in test positivity rates, as depicted in Figure 3B, were influenced by variations in the months during which the 2010, 2015, 2018, and 2021 surveys were conducted, we conducted a comparison of clusters sampled in the same months but surveyed in different years. The analysis involved 740 clusters surveyed in the months of October and November, with data spanning all four survey years, as well as December, with data available for three survey years. Specifically, when examining clusters sampled in October, it was observed that 51% of the clusters in the 2021 survey reported zero positive tests, in contrast to 53% in the 2018 survey, 50% in the 2015 survey, and 32% in the 2010 survey (Appendix A). For clusters sampled in November, the percentages of clusters reporting zero positive tests were 54% in both 2021 and 2018, compared to 46% in 2015 and 43% in 2010 (Appendix A). Likewise, for clusters sampled in December, 100% of the clusters in the 2021 survey had zero positive tests, whereas this figure was 66% for the 2018 survey and only 13% for the 2010 survey (Appendix A).

To evaluate whether the observed findings were affected by regional differences, including climate variations during the months of data collection, we analyzed and visualized malaria test positivity rates by geopolitical region for each survey month. According to Figure 4, except for the clusters sampled in December, the samples from October and November covered all geopolitical zones across all DHS/MIS survey years. While the year-over-year decline in median test positivity rates was not consistently linear, it is notable that the median rate in 2021 was substantially lower than the rates observed in 2010 in the months of October and November for all regions. However, the number of children tested for malaria varied annually. The 2021 survey featured the largest number of children tested for malaria per geopolitical region, particularly in October and December. It is uncertain whether these differences in the number of children tested account for the observed trend.

### 3.2. Identifying Predictors of Malaria Test Positivity and Visualizing Bivariate Associations to Inform Intervention Prioritization

Bivariate analysis provided insight into the unadjusted functional relationships between the number of malaria positives and all 26 potential risk factors. It played a crucial role in guiding variable selection for the multivariable regression, especially for highly correlated variables, defined as those exhibiting a correlation coefficient of 60% or higher. Visualizations of the individual covariate distributions, the outcomes of the correlation analysis, and the results of the bivariate analysis can be found in Appendix A.

The final multivariable prediction model for malaria test positivity, chosen based on the lowest AIC values among a series of 21 models with varying covariates, was a Poisson model. The model QQ plot demonstrated that the model predictions closely aligned with the Poisson distribution (see Appendix A). The selected final model incorporated the following covariates: percentage of individuals with post-primary education, percentage of individuals in the rich wealth quintiles, percentage of individuals residing in improved housing in 2015, all-age population density, median age, percentage of children under the age of five seeking medical treatment for fevers, total precipitation, and enhanced vegetation index. The subsequent sections present findings from both single-variable and multivariable models, elucidating how changes in these covariates impact malaria test positivity rates in the comprehensive DHS dataset spanning from 2010 to 2018 and 2021.

#### 3.2.1. Clusters with the Lowest Educational Attainment and Wealth Were at the Highest Risk of Malaria

Socioeconomic variables exhibited a negative association with malaria transmission intensity, although this effect appeared less pronounced and exhibited greater uncertainty in the multivariate analysis (Figure 5A,B). The malaria test positivity rate declined with increases in the percentage of individuals with post-primary education in both unadjusted and adjusted analyses. Notably, the malaria test positivity rate displayed a consistent decline with increasing percentages of individuals with post-primary education, evident in both unadjusted and adjusted analyses. However, it is worth noting that the impact of educational attainment appeared most robust when the percentage was below 50%, as illustrated in Figure 5. Similarly, reductions in malaria test positivity rates were observed with increasing percentages of individuals falling within the rich wealth quintiles, particularly within the ranges of 0 to 50% and 80 to 100%. This trend was consistent in both the adjusted and unadjusted analyses. Notably, clusters characterized by the lowest socioeconomic status exhibited the highest risk of malaria in both analyses.

#### 3.2.2. High Population Density and Younger Median Age Correlated with Higher Malaria Transmission Intensity

In the unadjusted analysis, the malaria test positivity rate displayed a decline with increasing all-age population density, up to a threshold of 8000 persons per square kilometer, after which the decline plateaued (Figure 6A). However, in the adjusted analysis, the malaria test positivity rate remained relatively stable up to 17,000 persons per square kilometer, beyond which an increase in malaria test positivity was observed, albeit with significant uncertainty (Figure 6B). Furthermore, reductions in malaria test positivity rates were evident with rising median age, particularly beyond a median age of 18 years old, as shown in the unadjusted model. However, it is noteworthy that the impact of median age on malaria positivity appeared to diminish in the adjusted analysis (Figure 6C,D).

#### 3.2.3. Higher Enhanced Vegetation Index Was Positively Associated with the U5 Malaria Test Positivity Rate

In the unadjusted analysis, there was a notable correlation between increasing the malaria test positivity rate and higher values of the enhanced vegetation index, which is indicative of vegetation cover and growth. The most substantial reductions in the malaria test positivity rate, although characterized by a high degree of uncertainty, were observed at approximate vegetation indices of 0.5 and 0.76 (as depicted in Figure 7A). However, it is important to note that the influence of the enhanced vegetation index on malaria test positivity was notably reduced in the adjusted analysis (refer to Figure 7B).

#### 3.2.4. Effects of Housing, Care Seeking, and Precipitation

In the unadjusted analysis, an increase in the proportion of individuals residing in improved housing in the year 2015 was associated with a reduction in malaria test positivity rates. However, in the adjusted analysis, these trends appeared relatively flat (Figure 8A,B). Variables used for the adjustment were the percentage of individuals with post-primary education, the percentage of individuals in the rich wealth quintiles, all-age population density, median age, the percentage of U5 children that sought medical treatment for fevers, total precipitation, and enhanced vegetation index. Examining the relationship between the proportion of children under the age of five seeking medical treatment for fever and malaria test positivity rates, it was evident that the highest test positivity rates in the unadjusted analysis were observed at approximately 25% (Figure 8C,D). For total precipitation, during the survey month, a consistent negative relationship was observed in both unadjusted and adjusted analyses for total precipitation (Figure 8E,F). Variables used to adjust total precipitation include the percentage of individuals with post-primary education, the percentage of individuals in the rich wealth quintiles, all-age population density, median age, the percentage of individuals living in improved housing in 2015, the percentage of U5 children that sought medical treatment for fevers, and enhanced vegetation index.

## 4. Discussion

We conducted an in-depth analysis using urban data from the most recent four years of Nigeria’s DHS/MIS surveys. Our primary objectives were to assess differences in the magnitude of the prevalence of malaria among children under the age of five over time and at the cluster and geopolitical levels, identify predictors at the community level, and create both unadjusted and adjusted effect plots. These analyses were undertaken with the dual purpose of guiding intervention prioritization in urban areas and gaining insights into the public health implications of the identified predictors.

The analysis of malaria test positivity rates among children under the age of five in urban areas revealed consistently low rates that have shown a declining trend over time. The south south and northeast geopolitical regions had the largest number of clusters where zero test positivity rates were observed. Notably, states with the highest number of urban clusters reporting zero test positivity rates included Lagos, Rivers, and Abia. Interestingly, prior research has documented low malaria infection rates, as determined by microscopy, in urban Lagos, with rates as low as 8% and 0.9% [45,46,47,48]. However, it remains somewhat unclear why lower test positivity rates were observed in the urban clusters of Rivers and Abia, especially when previous studies conducted in urban areas have indicated higher test positivity rates among the study populations [49,50,51]. It is important to note that two of these earlier studies were health facility studies, which did not differentiate participants based on whether their specific place of residence was urban or rural. Moreover, recruitment in the health facility surveys could have been biased toward sicker individuals likely to test positive for malaria, potentially contributing to this discrepancy.

The unadjusted fitted lines, which illustrate the relationship between the U5 malaria test positivity rate and various indicators, such as educational attainment, wealth, age distribution, vegetation cover, housing quality, and treatment-seeking behavior, suggest that communities at high risk for malaria in urban areas are characterized by lower educational attainment, poverty, younger residents, poorer housing quality, and infrequent fever treatment-seeking behavior. After adjustment, the impact of educational attainment and wealth is diminished, indicating that they could be explained by other factors included in the model, such as housing quality, median age, and environmental factors, like vegetation cover. Our results align with findings documented in the existing literature [52]. For example, in a comprehensive review of the factors associated with malaria transmission in urban areas across sub-Saharan Africa, Silvia and Marshall highlighted relevant studies demonstrating that poor housing, which increases exposure to mosquitoes, inadequate waste disposal, and urban agriculture are among the factors contributing to elevated malaria risk in urban settings [52]. Additionally, the increased risk of malaria infections among children is well-documented, consistent with our study’s findings that clusters with a higher proportion of children are more likely to exhibit a higher malaria burden [24].

Currently, the distribution of insecticide-treated bednets (ITNs) and the administration of seasonal malaria chemoprevention (SMC) through mass campaigns are the cornerstones of national malaria control and prevention efforts in Nigeria [53]. However, our study findings suggest that these tools alone may be insufficient to reduce the malaria burden in urban areas. ITNs primarily reduce mosquito exposure indoors, with protection limited to sleeping hours and effectiveness diminishing with the aging of the nets. SMC is administered only during the rainy season and relies on adherence to the full treatment course.

To effectively reduce the malaria burden in urban areas, it is essential to supplement these interventions with strategies that address malaria risk stemming from environmental and socioeconomic factors, including poor-quality housing. Interventions such as housing modification and larval source management have demonstrated success in reducing malaria infections. For example, window and door screening has been linked to a 62% reduction in malaria incidence in a study conducted in Ethiopia and a 16% reduction in malaria parasite prevalence, even in the absence of bednet usage [54]. Larval source management, involving habitat modification and manipulation, has shown the ability to reduce the density of adult mosquitoes [55]. Behavior change interventions, including educational campaigns and resources to support high-risk communities in understanding the drivers of malaria risk, can be instrumental. A combination of these interventions is likely to lead to a significant decrease in the malaria burden in urban areas.

The study findings offer valuable guidance on how to identify at-risk communities and prioritize them during resource allocation. At a regional scale, resources for urban malaria control could follow the order shown in Figure 2D, with a priority focus on the northwest and southwest regions. The information provided in Appendix A is also useful as it highlights states at high risk of malaria, such as Kebbi in the northwest and Ondo in the southwest. However, when determining intervention prioritization, population sizes in these regions and states must also be considered. Even areas with a lower burden may have a greater population at risk. To apply the insights gained from the multivariable models, employee data from major city employers can be utilized to identify areas where residents have lower levels of post-primary education. This information can then inform resource allocation, especially if local factors are believed to drive malaria transmission. Alternatively, it can guide the provision of prophylactic measures if transmission is predominantly influenced by mobility patterns. Furthermore, geospatial data, such as the enhanced vegetation index generated from high-resolution satellite imagery, can be employed to identify and prioritize high-risk areas. Nevertheless, it is crucial to emphasize that the meaningful application of these study methodologies for informing intervention prioritization and deprioritization decisions requires improved data that effectively capture the malaria burden, associated socio-demographic and environmental factors, and the geographic extent of urban areas, as well as local knowledge of major risk factors. Previous attempts to prioritize administrative units in Nigeria for ITN distribution, with the aim of addressing malaria risk, highlighted the challenges of distinguishing between high- and low-priority areas in the absence of high-quality data and a comprehensive understanding of the local context [56]. In addition, it is important to consider that predictive factors for malaria could vary based on the geographic scale of analysis [57].

This study faces certain limitations associated with data quality and availability. One notable limitation is the potentially low malaria test positivity rate observed in Nigeria’s northeast region, which may be attributed to under-sampling due to security concerns in that area. Additionally, it is important to recognize that DHS/MIS surveys typically provide data that may not fully represent urban settings at the state and zonal levels. Moreover, the timing of the surveys aligns with malaria transmission months in the southern geopolitical zones. As a result, drawing definitive conclusions regarding a low malaria burden across urban settings or making comparisons across geopolitical zones becomes challenging. To address potential biases related to temporal and regional variations when comparing malaria test positivity across survey years, we made efforts to mitigate this issue by comparing clusters sampled during the same months and in the same region. Fortunately, our findings from these analyses remained consistent, indicating a decline in the malaria burden over time. However, it is worth noting that these findings may be influenced by differences in the sampling strategy employed across the survey years. In addition, it is likely that the results of the 2021 survey may have been impacted by the SARS-CoV-2 pandemic, as it could have led to lower participation rates in the survey, decreased care-seeking behavior, and reduced access to interventions. The 2022 World Malaria Report lends support to the occurrence of disruptions in malaria services, such as bednet distribution through mass campaigns, and access to malaria diagnosis and treatment in 2021 [6].

Furthermore, it is essential to note that the reported test positivity rates do not account for potential changes in transmission that may occur over a 12-month period. Shifts in test positivity rates outside the months covered by the DHS/MIS surveys could lead to different conclusions regarding the burden and trends in malaria prevalence within urban areas. Additionally, the distribution of covariates derived from the DHS/MIS data is likely to be unrepresentative of urban settings. For instance, the measurement of educational attainment, as indicated by the percentage of individuals with post-primary education (secondary or college education), showed low values in most of the sampled clusters. In contrast, a greater proportion of clusters fell within the higher wealth quintiles or had improved housing infrastructure. If individuals accurately reported their educational attainment, it seems improbable that most clusters would fall within the higher wealth quintiles or have improved housing infrastructure, given the well-established positive correlation between educational attainment and these socioeconomic factors. [58,59]. We have made efforts to address these limitations by utilizing modeled covariates where possible. However, to provide a comprehensive understanding of the data and to guide future study or data collection efforts, we have included visualizations of these study covariates in Appendix A.

The distance displacement of clusters designed to protect the confidentiality of respondents meant that we could not examine the impact of the proximity of clusters on disease risk. Additionally, the lack of data on co-morbidities, such as HIV/AIDS and sickle cells, which increase the risk of developing severe malaria, meant that we could not adjust for them in the modeling analysis. In addition, due to data availability, our analysis is confined to children under the age of five years, whose malaria burden likely exhibits different transmission dynamics and determinants from older children and adults.

## 5. Conclusions

This study contributes to the existing research literature by examining variations in malaria among children under five (U5) in urban Nigeria and investigating associated risk factors. Improving the efficiency of intervention distribution has the potential to significantly reduce the malaria burden. As locally representative data become available for individual cities, the methodologies employed in this study can be easily adapted to inform intervention stratification strategies. Specifically, the identified predictive factors can help determine thresholds for prioritizing or de-prioritizing interventions, such as the distribution of bednets, in states and geopolitical regions with a high urban malaria burden. Our study underscores the importance of addressing environmental risk factors through housing improvements. With the ongoing trend of urbanization, there is a growing interest in tackling malaria within urban settings, as highlighted by the release of a framework by the WHO for responding to malaria in urban areas [60]. By shedding light on the existing data gaps within the Nigerian DHS/MIS, we aim to encourage increased investments aimed at enhancing both its design and usefulness. Additionally, we advocate for improvements in routine surveillance systems to further support malaria research and intervention efforts.

## Figures and Tables

**Figure 1 ijerph-21-00078-f001:**
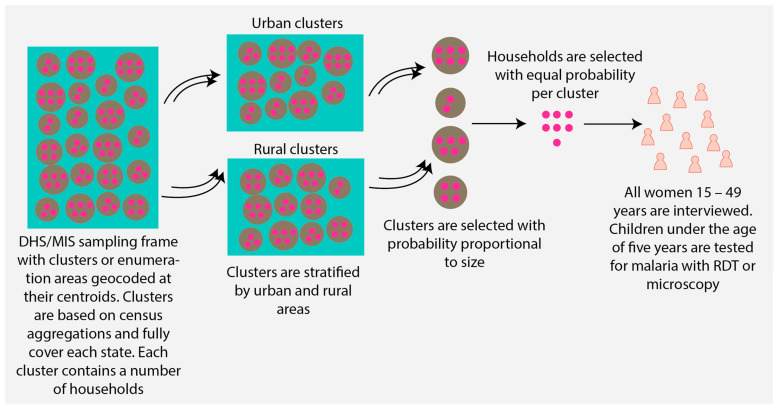
A typical sampling framework for a DHS/MIS survey in Nigeria illustrating the process of selecting study participants. Green rectangles denote the state level sampling frame, large brown circles represent clusters, and dark pink circles indicate households.

**Figure 2 ijerph-21-00078-f002:**
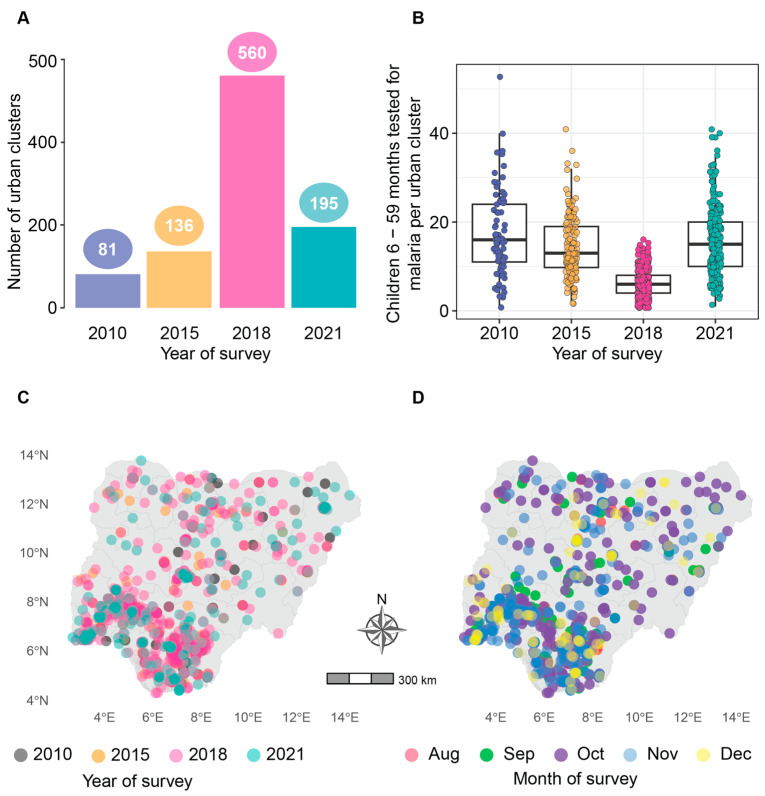
Urban clusters sampled in the 2010, 2015, 2018, and 2021 surveys: (**A**) number of clusters per survey year. Eighty-one clusters were sampled in 2010, 136 in 2015, 560 in 2018, and 195 in 2021; (**B**) number of children 6–59 months tested for malaria using microscopy per cluster and by year of survey; (**C**) cluster centroids mapped by year of survey within state-level administrative boundaries; and (**D**) cluster centroids mapped by month of survey within state-level administrative boundaries.

**Figure 3 ijerph-21-00078-f003:**
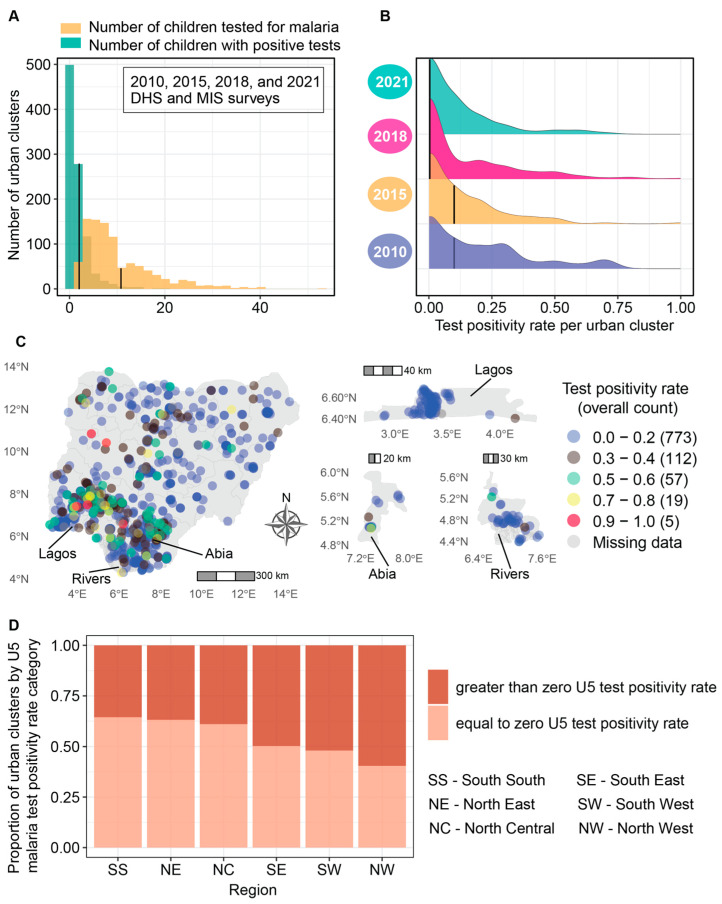
Distribution of malaria tests by microscopy, positive tests, and test positivity rates among children 6–59 months within urban clusters in the DHS/MIS 2010–2021: (**A**) distribution of the number of positive malaria tests (conducted by microscopy) per cluster (green); the mean = 2.9 (SD = 2.8) is depicted with a black line, and the distribution of the number tested for malaria per cluster (yellow); the mean = 10.2 (SD = 7.5) is depicted with a black line. Forty-nine percent (473) of the 972 clusters with non-missing values had zero positive tests. In all clusters, at least one child was tested for malaria; (**B**) density distribution of cluster test positivity rate by year of DHS survey. The thick black line is the median test positivity rate; (**C**) positive tests as a fraction of the number of children tested geolocated within state-level geographical boundaries. The majority of surveyed clusters in Lagos, Borno, and Akwa Ibom had a zero test positivity rate; (**D**) regional differences in the proportion of clusters at and above a zero malaria test positivity rate.

**Figure 4 ijerph-21-00078-f004:**
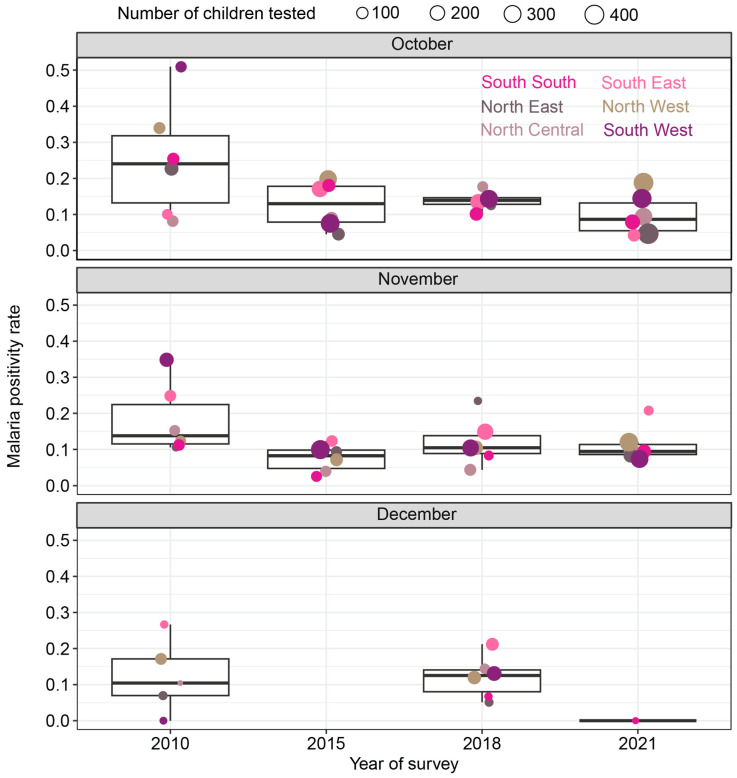
The malaria test positivity rate disaggregated by geopolitical region, year, and month of survey. Each point is sized by the number of children tested for malaria by microscopy.

**Figure 5 ijerph-21-00078-f005:**
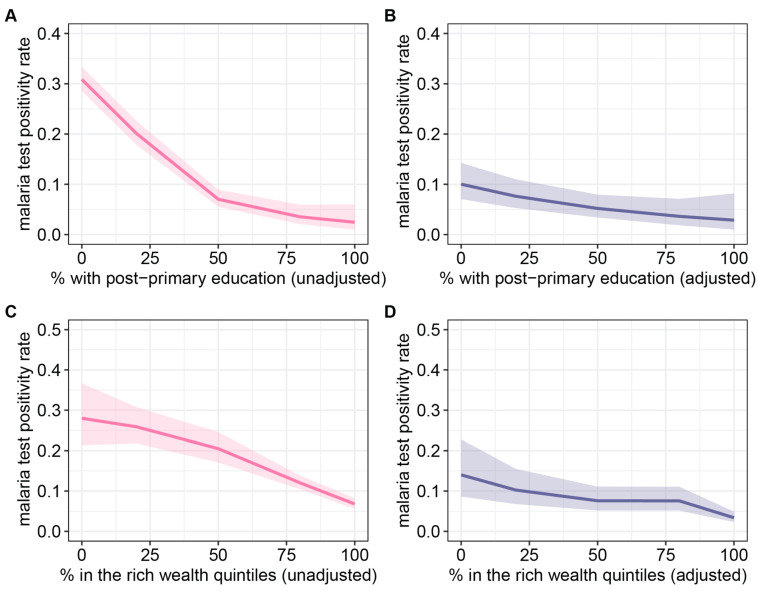
Effect plots of bivariate and multivariate regression analysis for indicators of educational attainment and wealth: (**A**) unadjusted and (**B**) adjusted effect of the percentage of individuals with post-primary education on the malaria test positivity rate. Percentage of individuals with post-primary education was adjusted for the percentage of individuals in the rich wealth quintiles, the percentage of individuals living in improved housing in 2015, all-age population density, median age, the percentage of children under the age of five that sought medical treatment for fevers, total precipitation, and enhanced vegetation index; (**C**) unadjusted and (**D**) adjusted effect of the percentage of individuals in the rich wealth quintiles on the malaria test positivity rate. The percentage of individuals in the rich wealth quintiles was adjusted for the percentage of individuals with post-primary education, the percentage of individuals living in improved housing in 2015, all-age population density, median age, the percentage of children under the age of five that sought medical treatment for fevers, total precipitation, and enhanced vegetation index.

**Figure 6 ijerph-21-00078-f006:**
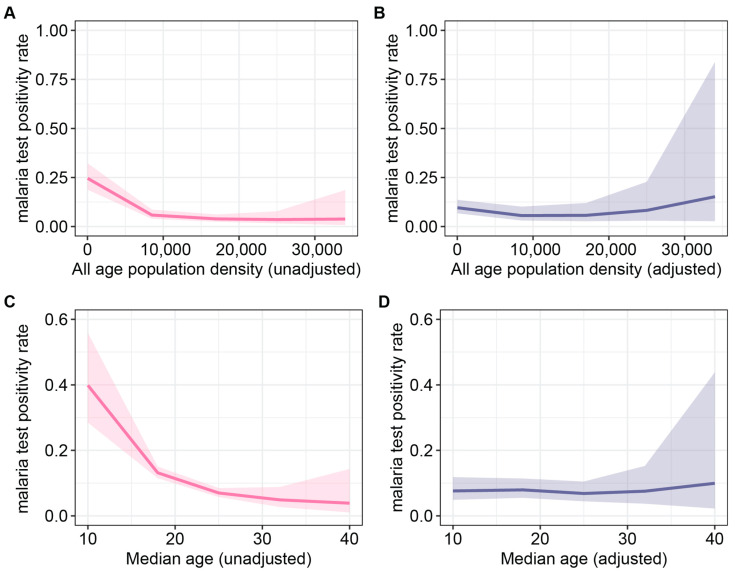
Effect plots of the bivariate and multivariate regression analysis for all-age population density and median age: (**A**) unadjusted and (**B**) adjusted effect of all-age population density (persons per square kilometer) on the malaria test positivity rate. All-age population density was adjusted for the percentage of individuals with post-primary education, the percentage of individuals in the rich wealth quintiles, the percentage of individuals living in improved housing in 2015, median age, the percentage of children under the age of five that sought medical treatment for fevers, total precipitation, and enhanced vegetation index; (**C**) unadjusted and (**D**) adjusted effect of median age in years on the malaria test positivity rate. Median age was adjusted for the percentage of individuals with post-primary education, the percentage of individuals in the rich wealth quintiles, all-age population density, the percentage of individuals living in improved housing in 2015, the percentage of children under the age of five that sought medical treatment for fevers, total precipitation, and enhanced vegetation index.

**Figure 7 ijerph-21-00078-f007:**
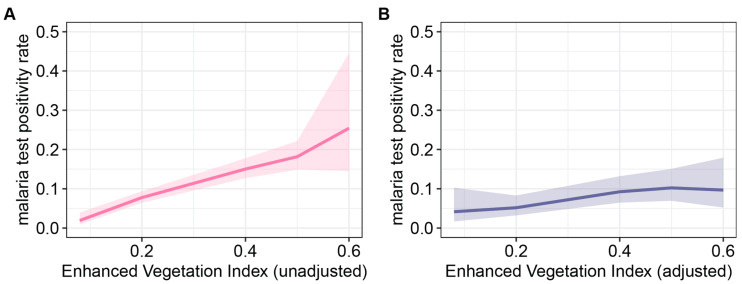
Effect plots of the bivariate and multivariate regression analysis for the enhanced vegetation index: (**A**) unadjusted and (**B**) adjusted effect of the enhanced vegetation index on the malaria test positivity rate. The enhanced vegetation index was adjusted for the percentage of individuals with post-primary education, the percentage of individuals in the rich wealth quintiles, the percentage of individuals living in improved housing in 2015, all-age population density, median age, the percentage of U5 children that sought medical treatment for fevers, and total precipitation.

**Figure 8 ijerph-21-00078-f008:**
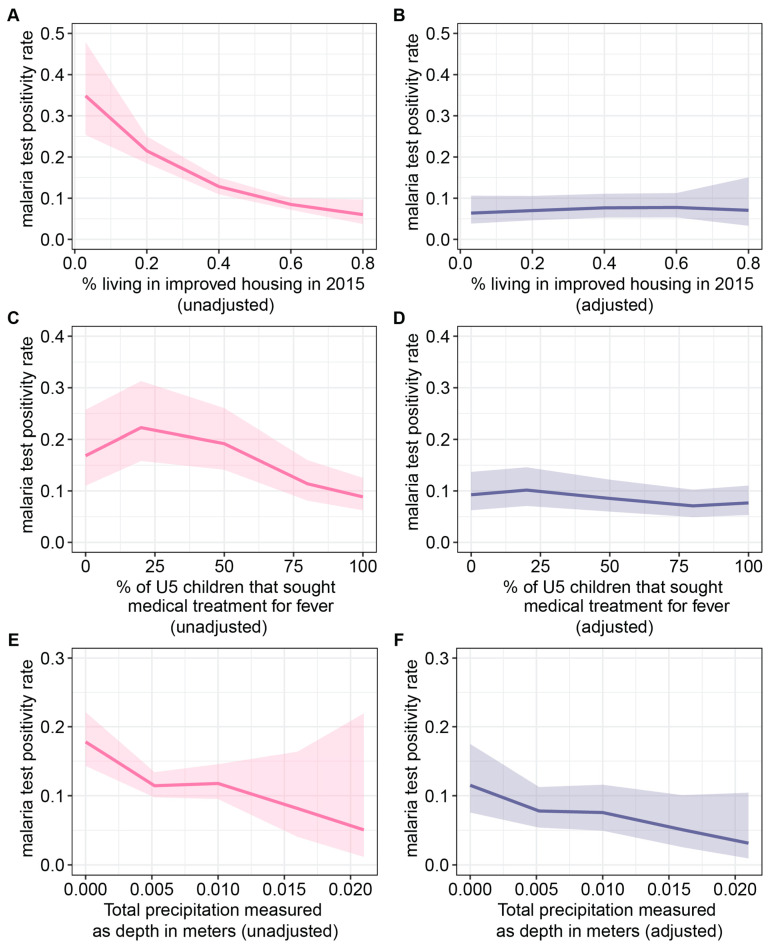
Effect plots of the bivariate and multivariate regression analysis for the percentage of individuals living in improved housing in 2015, the percentage of U5 children that sought medical treatment for fevers, and total precipitation: (**A**) unadjusted and (**B**) adjusted effect of the percentage of individuals living in improved housing in 2015; (**C**) unadjusted and (**D**) adjusted effect of the percentage of U5 children that sought medical treatment for fevers; (**E**) unadjusted and (**F**) adjusted effect of total precipitation in meters.

## Data Availability

Publicly available datasets were analyzed in this study. This DHS/MIS data can be obtained through the program website here: https://dhsprogram.com/. Sources for additional covariate data used in this work are provided in Table 1 of the main manuscript.

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
