# Peer review of "Socioeconomic, Demographic, and Environmental Factors May Inform Malaria Intervention Prioritization in Urban Nigeria"

_ijerph, 2024, doi:10.3390/ijerph21010078_

Round 1

Reviewer 1 Report

Comments and Suggestions for Authors

The manuscript presents a model to identify spatial and temporal variations in malaria burden. Comments follow.

-        Even though the manuscript addresses an important an interesting topic and it has potential, as it stands now, the contribution is not clear. The authors present their work as identifying spatial and temporal variations. However, there is no clear analysis of the spatial dimension. To start, there is no map that represents where the urban centers are located, there are no maps of where the clusters are within urban areas, there is nothing that describes where things area. This makes very difficult to visualize the spatial dimension. There is substantial amount of work that shows how this can be done. If indeed, spatial variations are uncovered, visualizing the underlying demographic factors would help as well. The authors mention they have the georeferenced data, so why are there no maps in the manuscript?

-        The abstract should mention the focus of analysis are children under 5 years of age. 

-        Ln. 62: “… 1km of streams travel to a rural area …”. Not sure what this means.

-        Ln. 68: “… intervention interventions.” Please revise.

-        Data section: the clusters are the key to the analysis. However, it is not clear how big they are, where exactly they are, there is no sense of the spatial dimension of these clusters. It seems these clusters are groupings of data. So perhaps, the label cluster is not accurate. When identifying spatial patterns, a cluster is a concentration of cases given a set of parameters. Therefore, using a cluster to define a level of data aggregation does not sound appropriate in this case. The authors need to clarify this.

-        Section 3.1.3: questions that are unanswered and unexplored include, to list a few: were the clusters in the same region over the months? Over the years? Were they close to each other? Were they in areas far apart?

-        Section 3.2.1: an interesting question that is raised when reading this section is: is a high educational attainment necessary for people to understand malaria risks? Or is formal education a proxy for living conditions? People who have low educational levels typically do not have the same opportunities as their counterpart, therefore their living conditions are not as good which results in a higher malaria risk. But even if people are uneducated, they can still understand educational campaigns against malaria and act accordingly. This is an important topic that the authors should address in their discussion.

-        Figure 7: has an extremely long caption for a figure.

-        Discussion section: this section needs to be strengthened and highlight the contributions of the work done.

-        Ln. 374: the paragraph was cut. Please revise.

-        Lns. 400-404: please revise sentence.

-        Conclusions:  first sentence. Please see my previous comment regarding the spatial component of the manuscript. The conclusion section also needs to be strengthened.

-        It is important for the authors to highlight how the results of their models can help the communities and the fight against malaria.

Author Response

We thank the reviewers for the time and effort taken to review our manuscript and provide in-depth and useful feedback. We have provided point-by-point responses below as well as revised the manuscript based on the reviewers’ recommendations. We are happy to provide additional communication and revisions if needed.

Reviewer Comments:

Reviewer 1

The manuscript presents a model to identify spatial and temporal variations in malaria burden. Comments follow.

-        Even though the manuscript addresses an important an interesting topic and it has potential, as it stands now, the contribution is not clear. The authors present their work as identifying spatial and temporal variations. However, there is no clear analysis of the spatial dimension. To start, there is no map that represents where the urban centers are located, there are no maps of where the clusters are within urban areas, there is nothing that describes where things area. This makes very difficult to visualize the spatial dimension. There is substantial amount of work that shows how this can be done. If indeed, spatial variations are uncovered, visualizing the underlying demographic factors would help as well. The authors mention they have the georeferenced data, so why are there no maps in the manuscript?

Response: We sincerely appreciate your thoughtful assessment and valuable feedback. In response to the concerns outlined above, we have implemented several revisions to address these issues.

Firstly, we have incorporated two descriptive maps in line 239, illustrating the locations of sampled urban clusters over different years and months. It's important to note that these maps are constrained to cluster circles, representing their centroids. This limitation arises because the DHS does not provide cluster boundaries. We have expanded the introduction to include this information and have also clarified our use of the term 'clusters.' In our context, clusters refer to aggregations of households, a terminology commonly employed by survey researchers familiar with the DHS/MIS. While we acknowledge the potential confusion due to its alternative usage in the context of disease outbreak aggregations, the term 'enumeration areas' is another option, albeit potentially confusing as well. We aim to mitigate any confusion by clearly defining the usage of 'clusters' in the introduction, anticipating that this will facilitate readers' understanding throughout the rest of the document.

In addressing the spatial variations under examination, we have added text to the introduction to explicitly state our objective: To delineate the geographic and temporal variations in the malaria test positivity rate among children under the age of five (U5) at both the cluster and geopolitical level (Lines 121 - 123). Our findings, particularly the visualization in Figure 2C depicting low malaria test positivity across most clusters, align with the outputs derived from this objective.

Furthermore, it is important to clarify that our analysis is currently limited to being descriptive, specifically for the malaria test positivity data. This limitation arises because further visualization of demographic data would not yield meaningful insights at the cluster and geospatial levels due to the absence of boundaries for urban areas and clusters, coupled with the displacement of clusters to safeguard confidentiality.

-        The abstract should mention the focus of analysis are children under 5 years of age. 

Response: Thank you, we have revised the abstract to reflect this suggestion.  Please see line 21.

-        Ln. 62: “… 1km of streams travel to a rural area …”. Not sure what this means.

Response: Thank you for catching this a error. A word was missing. We have revised sentence to read as follows “residing within a 1km distance from streams and travel to a rural area were statistically significant…” Please refer to line 75.

-        Ln. 68: “… intervention interventions.” Please revise.

Response: Many thanks for catching this error. The extra text has been deleted.

-        Data section: the clusters are the key to the analysis. However, it is not clear how big they are, where exactly they are, there is no sense of the spatial dimension of these clusters. It seems these clusters are groupings of data. So perhaps, the label cluster is not accurate. When identifying spatial patterns, a cluster is a concentration of cases given a set of parameters. Therefore, using a cluster to define a level of data aggregation does not sound appropriate in this case. The authors need to clarify this.

Response: Thank you for your valuable feedback. We have taken steps to clarify the usage of the term 'clusters' in the introduction, as detailed in our previous response. For the purposes of our analysis, we gauge the size of the clusters based on the number of children under the age of five (U5) tested for malaria per cluster, a metric highlighted in Figure 2b. This is because the DHS/MIS survey only provides information on the number of households and people per cluster but not on the spatial dimension of the clusters.

Figures 2c and 2d aim to geolocate the clusters in space and time. However, it's important to note that these visualizations are constrained due to the absence of boundaries for the clusters and the displacement of clusters for confidentiality reasons. These limitations have influenced our analysis strategy, focusing on the cluster and geopolitical levels, while providing some state-level information. It's also worth noting that the DHS/MIS program ensures that displaced clusters remain within state boundaries. Given that each state is associated with specific geopolitical units, we can correlate each cluster with its corresponding geopolitical region.

-        Section 3.1.3: questions that are unanswered and unexplored include, to list a few: were the clusters in the same region over the months? Over the years? Were they close to each other? Were they in areas far apart?

Response: Many thanks for these observations. We acknowledge that these are valid questions that will support contextualizing the findings. We have changed Figure 3 to illustrate that for the months of October and November, the clusters were drawn from all six geopolitical regions for each survey year analyzed. Our findings remain the same, that is malaria test positivity rate appeared to have declined since 2010. However, it’s possible that these results are an artefact of differences in survey design (more children appeared to have been sampled in later surveys). This limitation has been highlighted in the discussion (refer to lines 597 - 598).

Due to the distance displacement of the clusters to protect the confidentiality of the respondents, we cannot evaluate the proximity of the clusters to each other. We highlight this issue in lines 623 - 624.

-        Section 3.2.1: an interesting question that is raised when reading this section is: is a high educational attainment necessary for people to understand malaria risks? Or is formal education a proxy for living conditions? People who have low educational levels typically do not have the same opportunities as their counterpart, therefore their living conditions are not as good which results in a higher malaria risk. But even if people are uneducated, they can still understand educational campaigns against malaria and act accordingly. This is an important topic that the authors should address in their discussion.

Response: You have raised a significant and thought-provoking point regarding the potential relationship between malaria risk and educational attainment. Reviewing both the adjusted and unadjusted plots (Figure 4a – b), it becomes readily apparent that the influence of educational attainment diminishes considerably when we account for other modelled factors such as housing quality, age, and vegetation. This observation lends strong support to the hypothesis that educational attainment may serve as a proxy for environmental and demographic variables.

We wish to draw your attention to our acknowledgment of this point in the manuscript, specifically within lines 474 - 477. Additionally, we have provided citations to relevant literature in support of this argument, which can be found between lines 482 and 486. Furthermore, our discussion delves into the broader implications of these findings, as outlined in lines 501 - 535. It is noteworthy that our discussion is situated within the context of the currently established malaria interventions, namely, bednets and seasonal malaria chemoprevention. We posit that a comprehensive strategy combining these interventions with housing modifications, larval source management, and behavior change campaigns aimed at promoting their uptake and effective implementation holds significant promise in reducing the overall malaria burden.

-        Figure 7: has an extremely long caption for a figure.

Response: Many thanks for catching this, the caption has been shortened by transferring some of the content into the main text.

-        Discussion section: this section needs to be strengthened and highlight the contributions of the work done

Response: We thank you for suggesting this, we have added more content to the discussion as referenced in our response above

-        Ln. 374: the paragraph was cut. Please revise.

Response: We have reviewed the line referenced and can’t seem to identify where the paragraph was cut in the version of the discussion that we submitted. We have also extensively reviewed the discussion to ensure that all paragraphs are complete.

-        Lns. 400-404: please revise sentence.

Response: We have extensively revised the discussion to improve flow and clarity.

-        Conclusions:  first sentence. Please see my previous comment regarding the spatial component of the manuscript. The conclusion section also needs to be strengthened.

Response: We have addressed the previous comment on the spatial component and revised the conclusion. Many thanks for your suggestions.

-        It is important for the authors to highlight how the results of their models can help the communities and the fight against malaria.

Response: Many thanks for highlighting this issue. We have revised the discussion to include information on how this study findings can support communities in malaria control and prevention. Please refer to lines 516 - 551.

Reviewer 2 Report

Comments and Suggestions for Authors

Synthesis researchs such as the one presented in this manuscript: “Socioeconomic, demographic and environmental factors may inform malaria intervention prioritization in urban Nigeria” are extremely important in the context of public health and the development/improvement of public policies. The authors were very fearless in taking on the analysis of such a broad set of data collected from such varied realities of health services in the all country, but were successful in the analysis process, applying population, spatial and temporal adjustments. Another strong point of the study refers to the rich and clear presentation of descriptive data, often neglected in other papers.

I congratulate the authors and hope that they continue this line of research even more deeply. Below are some observations:

1- It may seem obvious, but I missed in Introduction a paragraph that briefly clarified age-epidemiological aspects of malaria that led the authors to select the children under the age of five years as the focus in this research.

2-line 68: “...intervention interventions...”

3-Among the adjustment variables, would the authors find it opportune to include co-morbidities? If not considered at this time, perhaps include a brief mention in the discussion.

4-The year 2021 showed trends that were clearly different from other years and this made me wonder if perhaps the SARS-CoV-2 pandemic had at least impacted data collection and exams. Regardless of its effect on research (which is difficult to verify), I think it would be appropriate in the discussion to mention something about the period and possible influences.

5-  Finally, if the authors find it interesting, I recommend a brief discussion of the impact of selecting the geographic scale in the analysis of health events, citing references such as Gracie et al., 2014 (Int. J. Environ. Res. Public Health 2014, 11, 10366-10383; doi:10.3390/ijerph111010366)

Author Response

Reviewer 2

Synthesis research such as the one presented in this manuscript: “Socioeconomic, demographic and environmental factors may inform malaria intervention prioritization in urban Nigeria” are extremely important in the context of public health and the development/improvement of public policies. The authors were very fearless in taking on the analysis of such a broad set of data collected from such varied realities of health services in the all country, but were successful in the analysis process, applying population, spatial and temporal adjustments. Another strong point of the study refers to the rich and clear presentation of descriptive data, often neglected in other papers.

I congratulate the authors and hope that they continue this line of research even more deeply. Below are some observations:

1- It may seem obvious, but I missed in Introduction a paragraph that briefly clarified age-epidemiological aspects of malaria that led the authors to select the children under the age of five years as the focus in this research.

Response:  Thank you very much for bringing this matter to our attention. Indeed, analyzing data across all age groups would be immensely beneficial. However, it's important to note that the DHS/MIS surveys focus specifically on malaria infections in children under five years of age. We have acknowledged this as a limitation in our analysis (refer to line 101 – 105 in the introduction and lines 628 – 631 in the discussion). Despite this, it is crucial to remember that this age group is central to malaria prevention and control efforts, regardless of seasonality and transmission intensity. Therefore, we believe that our findings retain their relevance and contribute meaningfully to this important field of study.

2-line 68: “...intervention interventions...”

Response: Thank you for catching this, we have revised

3-Among the adjustment variables, would the authors find it opportune to include co-morbidities? If not considered at this time, perhaps include a brief mention in the discussion.

Response: Great suggestion, we are unable to explore it now because it may be impossible to establish comorbidity using the DHS/MIS. We have included this as a limitation in lines 626 - 627.

4-The year 2021 showed trends that were clearly different from other years and this made me wonder if perhaps the SARS-CoV-2 pandemic had at least impacted data collection and exams. Regardless of its effect on research (which is difficult to verify), I think it would be appropriate in the discussion to mention something about the period and possible influences.

Response: Thank you for your observation, we have revised discussion to reflect this suggestion in lines 600 - 602.

5-  Finally, if the authors find it interesting, I recommend a brief discussion of the impact of selecting the geographic scale in the analysis of health events, citing references such as Gracie et al., 2014 (Int. J. Environ. Res. Public Health 2014, 11, 10366-10383; doi:10.3390/ijerph111010366)

Response: Thank you for recommending the paper, we have included a brief mention in the discussion (line 562 - 563)

Reviewer 3 Report

Comments and Suggestions for Authors

Introduction

Well written and provides relevant information on the context of the study.

Methods

Well described and appropriate.

Results

I am not sure I understand that scale in figure 3. Please could the authors explain what is meant by density on the Y axis.

Discussion

Limitations of the study and its data are identified. Although these limitations restrict the value of the work, the authors describe how their approach would be useful for generating prioritization thresholds for interventions.

Lines 424-441 Please could the authors provide some references to show where such prioritization has been achieved or discussed elsewhere? Modelling studies of this type are not new, so I wonder if any have allow for risk-informed interventions.

Minor edits

Line 51 use lowercase for species name

Line 61-62 check sentence structure

Line 68 remove repeated word

Line 142 provide reference for the glmmTMB package

Author Response

Reviewer 3

Introduction

Well written and provides relevant information on the context of the study.

Response: Thank you for reviewing our paper and letting us know that you find the information presented important.

Methods

Well described and appropriate.

Response: Thank you 

Results

I am not sure I understand that scale in figure 3. Please could the authors explain what is meant by density on the Y axis.

Response: Thank you for asking for us to explain the y-axis. We have added a description that the y-axis in the density plot is the probability density function for the kernel density estimation. We assume that the between the range of 0 and 1, that test positivity rate is continuous, enabling us to generate the plot. Based on feedback from another reviewer, the figure referenced has been placed in the supplement and is now Supplementary Figure 1. We have generated a new figure 3 to illustrate that clusters in the month of October and November arose from the same geopolitical region.

Discussion

Limitations of the study and its data are identified. Although these limitations restrict the value of the work, the authors describe how their approach would be useful for generating prioritization thresholds for interventions.

Response: Many thanks for your feedback

Lines 424-441 Please could the authors provide some references to show where such prioritization has been achieved or discussed elsewhere? Modelling studies of this type are not new, so I wonder if any have allow for risk-informed interventions.

Response: Many thanks for your question. We know of only one case where prioritization approaches  have been discussed in the context of malaria intervention planning. We have discussed and cited the paper in lines 559 - 562.

Minor edits

Line 51 use lowercase for species name

Line 61-62 check sentence structure

Line 68 remove repeated word

Line 142 provide reference for the glmmTMB package

Response: Many thanks for catching these, we have revised all accordingly.

Round 2

Reviewer 1 Report

Comments and Suggestions for Authors

Even though the authors tried to address the reviewer’s comments, there are still major concerns with the manuscript. As mentioned in the first review round, the manuscript does not present a proper spatial and temporal analysis of malaria incidence. As understood in the field spatial and spatio-temporal epidemiology, a true analysis would be something similar to what is done in https://www.ncbi.nlm.nih.gov/pmc/articles/PMC7584089/ (spatial analysis) or https://malariajournal.biomedcentral.com/articles/10.1186/s12936-023-04577-4 (spatio-temporal analysis), just to list two examples. If the dataset does not allow this kind of analysis, then it is recommended the authors remove from the manuscript any reference to spatio-temporal analysis and leave only the analysis that is not related to space or geography. Looking at the results section, the only discussion about space is regarding the locations where the incidence is high. This does not qualify as a spatial analysis.

As mentioned also in the first round of comments, this is a topic that is important and the analysis is valuable. I suggest the authors remove from the manuscript references to spatial and spatio-temporal analysis and present their work as an analysis of malaria incidence without focusing on space.

Specific Comments

-        It is important to provide an example of what an enumeration unit looks like. The authors explain how this is equivalent to a cluster, but it is still not clear when they use the term “cluster” what they are referring to.

-        Figure 1 c and d: the maps require work. They need to include a north arrow and a scale bar. Also, the color ramp in the legend needs to be revised to make the colors more distinct. They are too similar which makes very difficult to identify individual categories.

-        The same comment for Figure 2c.

Author Response

Response to Reviewer

We sincerely appreciate the detailed comments, valuable suggestions, and clear guidance provided by the Reviewer. In response, we have implemented the requested corrections, including removing references to spatial analysis. We are committed to maintaining high standards of quality and accuracy in our work and hope that these revisions meet the Reviewer's expectations. We remain open to any further feedback or suggestions that can enhance the quality of our manuscript. Once again, we thank the Reviewer for their constructive and insightful input.

Comments

 Even though the authors tried to address the reviewer’s comments, there are still major concerns with the manuscript. As mentioned in the first review round, the manuscript does not present a proper spatial and temporal analysis of malaria incidence. As understood in the field spatial and spatio-temporal epidemiology, a true analysis would be something similar to what is done in https://www.ncbi.nlm.nih.gov/pmc/articles/PMC7584089/ (spatial analysis) or https://malariajournal.biomedcentral.com/articles/10.1186/s12936-023-04577-4 (spatio-temporal analysis), just to list two examples. If the dataset does not allow this kind of analysis, then it is recommended the authors remove from the manuscript any reference to spatio-temporal analysis and leave only the analysis that is not related to space or geography. Looking at the results section, the only discussion about space is regarding the locations where the incidence is high. This does not qualify as a spatial analysis.

As mentioned also in the first round of comments, this is a topic that is important and the analysis is valuable. I suggest the authors remove from the manuscript references to spatial and spatio-temporal analysis and present their work as an analysis of malaria incidence without focusing on space.

Response: Thank you for your feedback. In response, we have made the following revisions to our manuscript:

  • Removed all references to spatial and spatial-temporal analysis to focus more on the core study objectives.
  • Refined the study aims to read as follows: (1) Describe the magnitude and variations in malaria prevalence among children under the age of five years (U5), analyzed at both the cluster and geopolitical levels in individual DHS/MIS surveys (see line 122).
  • Modified subtitles (line 22) and sections in the discussion (line 433) to reflect these changes and ensure consistency throughout the document."

Specific Comments

-        It is important to provide an example of what an enumeration unit looks like. The authors explain how this is equivalent to a cluster, but it is still not clear when they use the term “cluster” what they are referring to.

Response: Thank you very much for your valuable suggestion. In response, we have added a new figure (Figure 1) to the methods section, which illustrates the sampling framework used in the DHS/MIS surveys. This figure visually represents the enumeration areas or clusters, providing a clearer understanding of our methodology and the rationale behind our use of the term ‘cluster’. We believe this addition will enhance the reader's insight into the structure and approach of our study. Please feel free to share any further thoughts or questions you may have regarding this new inclusion.

-        Figure 1 c and d: the maps require work. They need to include a north arrow and a scale bar. Also, the color ramp in the legend needs to be revised to make the colors more distinct. They are too similar which makes very difficult to identify individual categories.

-        The same comment for Figure 2c.

Response: Many thanks for your valuable feedback. We have added the north arrow and scale bar for the maps in Figure 1c, 1 d and 2c. We We have also generated more distinct color ramps.